# Unsupervised Learning of Visual Features by Contrasting Cluster Assignments

**Mathilde Caron**[1,2]        **Ishan Misra**[2]        **Julien Mairal**[1]

**Priya Goyal**[2]        **Piotr Bojanowski**[2]        **Armand Joulin**[2]

[1] Inria*        [2] Facebook AI Research

## Abstract

Unsupervised image representations have significantly reduced the gap with supervised pretraining, notably with the recent achievements of contrastive learning methods. These contrastive methods typically work online and rely on a large number of explicit pairwise feature comparisons, which is computationally challenging. In this paper, we propose an online algorithm, SwAV, that takes advantage of contrastive methods without requiring to compute pairwise comparisons. Specifically, our method simultaneously clusters the data while enforcing consistency between cluster assignments produced for different augmentations (or "views") of the same image, instead of comparing features directly as in contrastive learning. Simply put, we use a "swapped" prediction mechanism where we predict the code of a view from the representation of another view. Our method can be trained with large and small batches and can scale to unlimited amounts of data. Compared to previous contrastive methods, our method is more memory efficient since it does not require a large memory bank or a special momentum network. In addition, we also propose a new data augmentation strategy, `multi-crop`, that uses a mix of views with different resolutions in place of two full-resolution views, without increasing the memory or compute requirements. We validate our findings by achieving 75.3% top-1 accuracy on ImageNet with ResNet-50, as well as surpassing supervised pretraining on all the considered transfer tasks.

## 1   Introduction

Unsupervised visual representation learning, or self-supervised learning, aims at obtaining features without using manual annotations and is rapidly closing the performance gap with supervised pretraining in computer vision [9, 20, 37]. Many recent state-of-the-art methods build upon the instance discrimination task that considers each image of the dataset (or "instance") and its transformations as a separate class [15]. This task yields representations that are able to discriminate between different images, while achieving some invariance to image transformations. Recent self-supervised methods that use instance discrimination rely on a combination of two elements: (i) a contrastive loss [19] and (ii) a set of image transformations. The contrastive loss removes the notion of instance classes by directly comparing image features while the image transformations define the invariances encoded in the features. Both elements are essential to the quality of the resulting networks [9, 37] and our work improves upon both the objective function and the transformations.

Correspondence to `mathilde@fb.com`
Code: `https://github.com/facebookresearch/swav`

The contrastive loss explicitly compares pairs of image representations to push away representations from different images while pulling together those from transformations, or views, of the same image. Since computing all the pairwise comparisons on a large dataset is not practical, most implementations approximate the loss by reducing the number of comparisons to random subsets of images during training [9, 20, 49]. An alternative to approximate the loss is to approximate the task—that is to relax the instance discrimination problem. For example, clustering-based methods discriminate between groups of images with similar features instead of individual images [6]. The objective in clustering is tractable, but it does not scale well with the dataset as it requires a pass over the entire dataset to form image "codes" (*i.e.*, cluster assignments) that are used as targets during training. In this work, we use a different paradigm and propose to compute the codes online while enforcing consistency between codes obtained from views of the same image. Comparing cluster assignments allows to contrast different image views while not relying on explicit pairwise feature comparisons. Specifically, we propose a simple "swapped" prediction problem where we predict the code of a view from the representation of another view. We learn features by **Sw**apping **A**ssignments between multiple **V**iews of the same image (**SwAV**). The features and the codes are learned online, allowing our method to scale to potentially unlimited amounts of data. In addition, SwAV works with small and large batch sizes and does not need a large memory bank [49] or a momentum encoder [20].

Besides our online clustering-based method, we also propose an improvement to the image transformations. Most contrastive methods compare one pair of transformations per image, even though there is evidence that comparing more views during training improves the resulting model [37]. In this work, we propose multi-crop that uses smaller-sized images to increase the number of views while not increasing the memory or computational requirements during training. We also observe that mapping small parts of a scene to more global views significantly boosts the performance. Directly working with downsized images introduces a bias in the features [45], which can be avoided by using a mix of different sizes. Our strategy is simple, yet effective, and can be applied to many self-supervised methods with consistent gain in performance.

We validate our contributions by evaluating our method on several standard self-supervised benchmarks. In particular, on the ImageNet linear evaluation protocol, we reach 75.3% top-1 accuracy with a standard ResNet-50, and 78.5% with a wider model. We also show that our multi-crop strategy is general, and improves the performance of different self-supervised methods, namely SimCLR [9], DeepCluster [6], and SeLa [2], between 2% and 4% top-1 accuracy on ImageNet. Overall, we make the following contributions:

- We propose a scalable online clustering loss that improves performance by +2% on ImageNet and works in both large and small batch settings without a large memory bank or a momentum encoder.

- We introduce the multi-crop strategy to increase the number of views of an image with no computational or memory overhead. We observe a consistent improvement of between 2% and 4% on ImageNet with this strategy on several self-supervised methods.

- Combining both technical contributions into a single model, we improve the performance of self-supervised by +4.2% on ImageNet with a standard ResNet and outperforms supervised ImageNet pretraining on multiple downstream tasks. This is the first method to do so without finetuning the features, *i.e.*, only with a linear classifier on top of frozen features.

## 2   Related Work

**Instance and contrastive learning.**   Instance-level classification considers each image in a dataset as its own class [4, 15, 49]. Dosovitskiy *et al*. [15] assign a class explicitly to each image and learn a linear classifier with as many classes as images in the dataset. As this approach becomes quickly intractable, Wu *et al*. [49] mitigate this issue by replacing the classifier with a memory bank that stores previously-computed representations. They rely on noise contrastive estimation [18] to compare instances, which is a special form of contrastive learning [24, 40]. He *et al*. [20] improve the training of contrastive methods by storing representations from a momentum encoder instead of the trained network. More recently, Chen *et al*. [9] show that the memory bank can be entirely replaced with the elements from the same batch if the batch is large enough. In contrast to this line of works, we avoid comparing every pair of images by mapping the image features to a set of trainable prototype vectors.

**Clustering for deep representation learning.** Our work is also related to clustering-based methods [2, 3, 6, 7, 17, 25, 50, 53, 54, 59]. Caron *et al.* [6] show that $k$-means assignments can be used as pseudo-labels to learn visual representations. This method scales to large uncurated dataset and can be used for pre-training of supervised networks [7]. However, their formulation is not principled and recently, Asano *et al.* [2] show how to cast the pseudo-label assignment problem as an instance of the optimal transport problem. We consider a similar formulation to map representations to prototype vectors, but unlike [2] we keep the soft assignment produced by the Sinkhorn-Knopp algorithm [12] instead of approximating it into a hard assignment. Besides, unlike Caron *et al.* [6, 7] and Asano *et al.* [2], we obtain online assignments which allows our method to scale gracefully to any dataset size.

**Handcrafted pretext tasks.** Many self-supervised methods manipulate the input data to extract a supervised signal in the form of a pretext task [1, 13, 26, 28, 30, 36, 38, 41, 42, 47, 48, 57]. We refer the reader to Jing *et al.* [27] for an exhaustive and detailed review of this literature. Of particular interest, Misra and van der Maaten [37] propose to encode the jigsaw puzzle task [39] as an invariant for contrastive learning. Jigsaw tiles are non-overlapping crops with small resolution that cover only part ($\sim$20%) of the entire image area. In contrast, our `multi-crop` strategy consists in simply sampling multiple random crops with two different sizes: a standard size and a smaller one.

## 3 Method

Our goal is to learn visual features in an online fashion without supervision. To that effect, we propose an online clustering-based self-supervised method. Typical clustering-based methods [2, 6] are offline in the sense that they alternate between a cluster assignment step where image features of the entire dataset are clustered, and a training step where the cluster assignments, *i.e.*, "codes" are predicted for different image views. Unfortunately, these methods are not suitable for online learning as they require multiple passes over the dataset to compute the image features necessary for clustering. In this section, we describe an alternative where we enforce consistency between codes from different augmentations of the same image. This solution is inspired by contrastive instance learning [49] as we do not consider the codes as a target, but only enforce consistent mapping between views of the same image. Our method can be interpreted as a way of contrasting between multiple image views by comparing their cluster assignments instead of their features.

More precisely, we compute a code from an augmented version of the image and predict this code from other augmented versions of the same image. Given two image features $\mathbf{z}_t$ and $\mathbf{z}_s$ from two different augmentations of the same image, we compute their codes $\mathbf{q}_t$ and $\mathbf{q}_s$ by matching these features to a set of $K$ prototypes $\{\mathbf{c}_1, \ldots, \mathbf{c}_K\}$. We then setup a "swapped" prediction problem with the following loss function:

$$L(\mathbf{z}_t, \mathbf{z}_s) \quad = \quad \ell(\mathbf{z}_t, \mathbf{q}_s) + \ell(\mathbf{z}_s, \mathbf{q}_t), \tag{1}$$

where the function $\ell(\mathbf{z}, \mathbf{q})$ measures the fit between features $\mathbf{z}$ and a code $\mathbf{q}$, as detailed later. Intuitively, our method compares the features $\mathbf{z}_t$ and $\mathbf{z}_s$ using the intermediate codes $\mathbf{q}_t$ and $\mathbf{q}_s$. If these two features capture the same information, it should be possible to predict the code from the other feature. A similar comparison appears in contrastive learning where features are compared directly [49]. In Fig. 1, we illustrate the relation between contrastive learning and our method.

### 3.1 Online clustering

Each image $\mathbf{x}_n$ is transformed into an augmented view $\mathbf{x}_{nt}$ by applying a transformation $t$ sampled from the set $\mathcal{T}$ of image transformations. The augmented view is mapped to a vector representation by applying a non-linear mapping $f_\theta$ to $\mathbf{x}_{nt}$. The feature is then projected to the unit sphere, *i.e.*, $\mathbf{z}_{nt} = f_\theta(\mathbf{x}_{nt})/\|f_\theta(\mathbf{x}_{nt})\|_2$. We then compute a code $\mathbf{q}_{nt}$ from this feature by mapping $\mathbf{z}_{nt}$ to a set of $K$ trainable prototypes vectors, $\{\mathbf{c}_1, \ldots, \mathbf{c}_K\}$. We denote by $\mathbf{C}$ the matrix whose columns are the $\mathbf{c}_1, \ldots, \mathbf{c}_k$. We now describe how to compute these codes and update the prototypes online.

**Swapped prediction problem.** The loss function in Eq. (1) has two terms that setup the "swapped" prediction problem of predicting the code $\mathbf{q}_t$ from the feature $\mathbf{z}_s$, and $\mathbf{q}_s$ from $\mathbf{z}_t$. Each term represents the cross entropy loss between the code and the probability obtained by taking a softmax of the dot

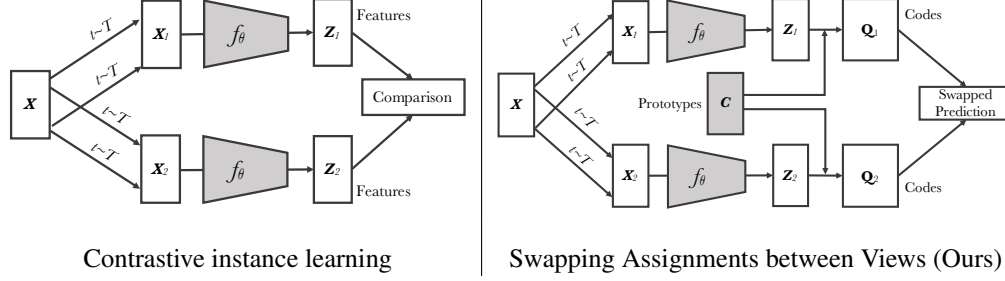

Contrastive instance learning | Swapping Assignments between Views (Ours)

Figure 1: **Contrastive instance learning (left) *vs*. SwAV (right).** In contrastive learning methods applied to instance classification, the features from different transformations of the same images are compared directly to each other. In SwAV, we first obtain "codes" by assigning features to prototype vectors. We then solve a "swapped" prediction problem wherein the codes obtained from one data augmented view are predicted using the other view. Thus, SwAV does not directly compare image features. Prototype vectors are learned along with the ConvNet parameters by backpropragation.

products of $\mathbf{z}_i$ and all prototypes in $\mathbf{C}$, *i.e.*,

$$\ell(\mathbf{z}_t, \mathbf{q}_s) = -\sum_k \mathbf{q}_s^{(k)} \log \mathbf{p}_t^{(k)}, \quad \text{where} \quad \mathbf{p}_t^{(k)} = \frac{\exp\left(\frac{1}{\tau}\mathbf{z}_t^\top \mathbf{c}_k\right)}{\sum_{k'} \exp\left(\frac{1}{\tau}\mathbf{z}_t^\top \mathbf{c}_{k'}\right)}. \tag{2}$$

where $\tau$ is a temperature parameter [49]. Taking this loss over all the images and pairs of data augmentations leads to the following loss function for the swapped prediction problem:

$$-\frac{1}{N}\sum_{n=1}^N \sum_{s,t\sim\mathcal{T}} \left[ \frac{1}{\tau}\mathbf{z}_{nt}^\top \mathbf{C}\mathbf{q}_{ns} + \frac{1}{\tau}\mathbf{z}_{ns}^\top \mathbf{C}\mathbf{q}_{nt} - \log\sum_{k=1}^K \exp\left(\frac{\mathbf{z}_{nt}^\top \mathbf{c}_k}{\tau}\right) - \log\sum_{k=1}^K \exp\left(\frac{\mathbf{z}_{ns}^\top \mathbf{c}_k}{\tau}\right) \right].$$

This loss function is jointly minimized with respect to the prototypes $\mathbf{C}$ and the parameters $\theta$ of the image encoder $f_\theta$ used to produce the features $(\mathbf{z}_{nt})_{n,t}$.

**Computing codes online.** In order to make our method online, we compute the codes using only the image features within a batch. Intuitively, as the prototypes $\mathbf{C}$ are used across different batches, SwAV clusters multiple instances to the prototypes. We compute codes using the prototypes $\mathbf{C}$ such that all the examples in a batch are equally partitioned by the prototypes. This equipartition constraint ensures that the codes for different images in a batch are distinct, thus preventing the trivial solution where every image has the same code. Given $B$ feature vectors $\mathbf{Z} = [\mathbf{z}_1, \ldots, \mathbf{z}_B]$, we are interested in mapping them to the prototypes $\mathbf{C} = [\mathbf{c}_1, \ldots, \mathbf{c}_K]$. We denote this mapping or codes by $\mathbf{Q} = [\mathbf{q}_1, \ldots, \mathbf{q}_B]$, and optimize $\mathbf{Q}$ to maximize the similarity between the features and the prototypes , *i.e.*,

$$\max_{\mathbf{Q}\in\mathcal{Q}} \text{Tr}\left(\mathbf{Q}^\top \mathbf{C}^\top \mathbf{Z}\right) + \varepsilon H(\mathbf{Q}), \tag{3}$$

where $H$ is the entropy function, $H(\mathbf{Q}) = -\sum_{ij}\mathbf{Q}_{ij}\log\mathbf{Q}_{ij}$ and $\varepsilon$ is a parameter that controls the smoothness of the mapping. We observe that a strong entropy regularization (*i.e.* using a high $\varepsilon$) generally leads to a trivial solution where all samples collapse into an unique representation and are all assigned uniformly to all prototypes. Hence, in practice we keep $\varepsilon$ low. Asano *et al*. [2] enforce an equal partition by constraining the matrix $\mathbf{Q}$ to belong to the transportation polytope. They work on the full dataset, and we propose to adapt their solution to work on minibatches by restricting the transportation polytope to the minibatch:

$$\mathcal{Q} = \left\{ \mathbf{Q} \in \mathbb{R}_+^{K\times B} \mid \mathbf{Q}\mathbf{1}_B = \frac{1}{K}\mathbf{1}_K, \mathbf{Q}^\top\mathbf{1}_K = \frac{1}{B}\mathbf{1}_B \right\}, \tag{4}$$

where $\mathbf{1}_K$ denotes the vector of ones in dimension $K$. These constraints enforce that on average each prototype is selected at least $\frac{B}{K}$ times in the batch.

Once a continuous solution $\mathbf{Q}^*$ to Prob. (3) is found, a discrete code can be obtained by using a rounding procedure [2]. Empirically, we found that discrete codes work well when computing codes in an offline manner on the full dataset as in Asano *et al*. [2]. However, in the online setting where

| Method | Arch. | Param. | Top1 |
|---|---|---|---|
| Supervised | R50 | 24 | 76.5 |
| Colorization [56] | R50 | 24 | 39.6 |
| Jigsaw [39] | R50 | 24 | 45.7 |
| NPID [49] | R50 | 24 | 54.0 |
| BigBiGAN [14] | R50 | 24 | 56.6 |
| LA [59] | R50 | 24 | 58.8 |
| NPID++ [37] | R50 | 24 | 59.0 |
| MoCo [20] | R50 | 24 | 60.6 |
| SeLa [2] | R50 | 24 | 61.5 |
| PIRL [37] | R50 | 24 | 63.6 |
| CPC v2 [23] | R50 | 24 | 63.8 |
| PCL [31] | R50 | 24 | 65.9 |
| SimCLR [9] | R50 | 24 | 70.0 |
| MoCov2 [10] | R50 | 24 | 71.1 |
| SwAV | R50 | 24 | **75.3** |

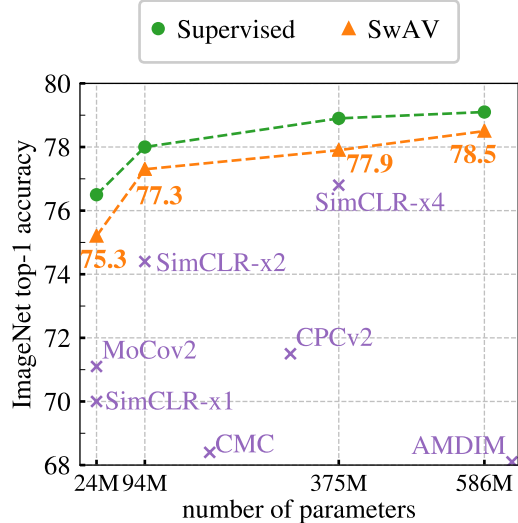

**Figure 2: Linear classification on ImageNet.** Top-1 accuracy for linear models trained on frozen features from different self-supervised methods. **(left)** Performance with a standard ResNet-50. **(right)** Performance as we multiply the width of a ResNet-50 by a factor $\times 2$, $\times 4$, and $\times 5$.

we use only minibatches, using the discrete codes performs worse than using the continuous codes. An explanation is that the rounding needed to obtain discrete codes is a more aggressive optimization step than gradient updates. While it makes the model converge rapidly, it leads to a worse solution. We thus preserve the soft code $\mathbf{Q}^*$ instead of rounding it. These soft codes $\mathbf{Q}^*$ are the solution of Prob. (3) over the set $\mathcal{Q}$ and takes the form of a normalized exponential matrix [12]:

$$\mathbf{Q}^* = \mathrm{Diag}(\mathbf{u}) \exp\left(\frac{\mathbf{C}^\top \mathbf{Z}}{\varepsilon}\right) \mathrm{Diag}(\mathbf{v}), \tag{5}$$

where $\mathbf{u}$ and $\mathbf{v}$ are renormalization vectors in $\mathbb{R}^K$ and $\mathbb{R}^B$ respectively. The renormalization vectors are computed using a small number of matrix multiplications using the iterative Sinkhorn-Knopp algorithm [12]. In practice, we observe that using only 3 iterations is fast and sufficient to obtain good performance. Indeed, this algorithm can be efficiently implemented on GPU, and the alignment of 4K features to 3K codes takes 35ms in our experiments, see § 4.

**Working with small batches.** When the number $B$ of batch features is too small compared to the number of prototypes $K$, it is impossible to equally partition the batch into the $K$ prototypes. Therefore, when working with small batches, we use features from the previous batches to augment the size of $\mathbf{Z}$ in Prob. (3). Then, we only use the codes of the batch features in our training loss. In practice, we store around 3K features, *i.e.*, in the same range as the number of code vectors. This means that we only keep features from the last 15 batches with a batch size of 256, while contrastive methods typically need to store the last 65K instances obtained from the last 250 batches [20].

### 3.2 Multi-crop: Augmenting views with smaller images

As noted in prior works [9, 37], comparing random crops of an image plays a central role by capturing information in terms of relations between parts of a scene or an object. Unfortunately, increasing the number of crops or "views" quadratically increases the memory and compute requirements. We propose a `multi-crop` strategy where we use two standard resolution crops and sample $V$ additional low resolution crops that cover only small parts of the image. Using low resolution images ensures only a small increase in the compute cost. Specifically, we generalize the loss of Eq (1):

$$L(\mathbf{z}_{t_1}, \mathbf{z}_{t_2}, \ldots, \mathbf{z}_{t_{V+2}}) = \sum_{i \in \{1,2\}} \sum_{v=1}^{V+2} \mathbf{1}_{v \neq i} \ell(\mathbf{z}_{t_v}, \mathbf{q}_{t_i}). \tag{6}$$

Note that we compute codes using only the full resolution crops. Indeed, computing codes for all crops increases the computational time and we observe in practice that it also alters the transfer performance

Table 1: **Semi-supervised learning on ImageNet with a ResNet-50.** We finetune the model with 1% and 10% labels and report top-1 and top-5 accuracies. *: *uses RandAugment [11].*

| | Method | 1% labels | | 10% labels | |
|---|---|---|---|---|---|
| | | Top-1 | Top-5 | Top-1 | Top-5 |
| | Supervised | 25.4 | 48.4 | 56.4 | 80.4 |
| *Methods using label-propagation* | UDA [51] | - | - | 68.8* | 88.5* |
| | FixMatch [44] | - | - | **71.5*** | 89.1* |
| *Methods using self-supervision only* | PIRL [37] | 30.7 | 57.2 | 60.4 | 83.8 |
| | PCL [31] | - | 75.6 | - | 86.2 |
| | SimCLR [9] | 48.3 | 75.5 | 65.6 | 87.8 |
| | SwAV | **53.9** | **78.5** | 70.2 | **89.9** |

of the resulting network. An explanation is that using only partial information (small crops cover only small area of images) degrades the assignment quality. Figure 3 shows that `multi-crop` improves the performance of several self-supervised methods and is a promising augmentation strategy.

## 4 Main Results

We analyze the features learned by SwAV by transfer learning on multiple datasets. We implement in SwAV the improvements used in SimCLR, *i.e.*, LARS [55], cosine learning rate [34, 37] and the MLP projection head [9]. We provide the full details and hyperparameters for pretraining and transfer learning in the supplementary material.

### 4.1 Evaluating the unsupervised features on ImageNet

We evaluate the features of a ResNet-50 [22] trained with SwAV on ImageNet by two experiments: linear classification on frozen features and semi-supervised learning by finetuning with few labels. When using frozen features (Fig. 2 left), SwAV outperforms the state of the art by $+4.2\%$ top-1 accuracy and is only $1.2\%$ below the performance of a fully supervised model. Note that we train SwAV during 800 epochs with large batches (4096). We refer to Fig. 3 for results with shorter trainings and to Table 3 for experiments with small batches. On semi-supervised learning (Table 1), SwAV outperforms other self-supervised methods and is on par with state-of-the-art semi-supervised models [44], despite the fact that SwAV is not specifically designed for semi-supervised learning.

**Variants of ResNet-50.** Figure 2 (right) shows the performance of multiple variants of ResNet-50 with different widths [29]. The performance of our model increases with the width of the model, and follows a similar trend to the one obtained with supervised learning. When compared with concurrent work like SimCLR, we see that SwAV reduces the difference with supervised models even further. Indeed, for large architectures, our method shrinks the gap with supervised training to $0.6\%$.

Table 2: **Transfer learning on downstream tasks.** Comparison between features from ResNet-50 trained on ImageNet with SwAV or supervised learning. We consider two settings. (1) Linear classification on top of frozen features. We report top-1 accuracy on all datasets except VOC07 where we report mAP. (2) Object detection with finetuned features on VOC07+12 `trainval` using Faster R-CNN [43] and on COCO [32] using DETR [5]. We report the most standard detection metrics for these datasets: $AP_{50}$ on VOC07+12 and AP on COCO.

| | Linear Classification | | | Object Detection | |
|---|---|---|---|---|---|
| | Places205 | VOC07 | iNat18 | VOC07+12 (Faster R-CNN) | COCO (DETR) |
| Supervised | 53.2 | 87.5 | 46.7 | 81.3 | 40.8 |
| SwAV | **56.7** | **88.9** | **48.6** | **82.6** | **42.1** |

Table 3: **Training in small batch setting.** Top-1 accuracy on ImageNet with a linear classifier trained on top of frozen features from a ResNet-50. All methods are trained with a batch size of 256. We also report the number of stored features, the type of cropping used and the number of epochs.

| Method | Mom. Encoder | Stored Features | multi-crop | epoch | batch | Top-1 |
|--------|:---:|:---:|:---:|:---:|:---:|:---:|
| SimCLR |   | 0 | $2\times224$ | 200 | 256 | 61.9 |
| MoCov2 | ✓ | $65,536$ | $2\times224$ | 200 | 256 | 67.5 |
| MoCov2 | ✓ | $65,536$ | $2\times224$ | 800 | 256 | 71.1 |
| SwAV |   | $3,840$ | $2\times160 + 4\times96$ | 200 | 256 | 72.0 |
| SwAV |   | $3,840$ | $2\times224 + 6\times96$ | 200 | 256 | 72.7 |
| SwAV |   | $3,840$ | $2\times224 + 6\times96$ | 400 | 256 | **74.3** |

## 4.2 Transferring unsupervised features to downstream tasks

We test the generalization of ResNet-50 features trained with SwAV on ImageNet (without labels) by transferring to several downstream vision tasks. In Table 2, we compare the performance of SwAV features with ImageNet supervised pretraining. First, we report the linear classification performance on the Places205 [58], VOC07 [16], and iNaturalist2018 [46] datasets. Our method outperforms supervised features on all three datasets. Note that SwAV is the first self-supervised method to surpass ImageNet supervised features on these datasets. Second, we report network finetuning on object detection on VOC07+12 using Faster R-CNN [43] and on COCO [32] with DETR [5]. DETR is a recent object detection framework that reaches competitive performance with Faster R-CNN while being conceptually simpler and trainable end-to-end. We use DETR because, unlike Faster R-CNN [21], using a pretrained backbone in this framework is crucial to obtain good results compared to training from scratch [5]. In Table 2, we show that SwAV outperforms the supervised pretrained model on both VOC07+12 and COCO datasets. Note that this is line with previous works that also show that self-supervision can outperform supervised pretraining on object detection [17, 20, 37]. We report more detection evaluation metrics and results from other self-supervised methods in the supplementary material. Overall, our SwAV ResNet-50 model surpasses supervised ImageNet pretraining on all the considered transfer tasks and datasets. We have released this model so other researchers might also benefit by replacing the ImageNet supervised network with our model.

## 4.3 Training with small batches

We train SwAV with small batches of 256 images on 4 GPUs and compare with MoCov2 and SimCLR trained in the same setup. In Table 3, we see that SwAV maintains state-of-the-art performance even when trained in the small batch setting. Note that SwAV only stores a queue of $3,840$ features. In comparison, to obtain good performance, MoCov2 needs to store $65,536$ features while keeping an additional momentum encoder network. When SwAV is trained using $2\times160 + 4\times96$ crops, SwAV has a running time $1.2\times$ higher than SimCLR with $2\times224$ crops and is around $1.4\times$ slower than MoCov2 due to the additional back-propagation [10]. Hence, one epoch of MoCov2 or SimCLR is faster in wall clock time than one of SwAV, but these methods need more epochs for good downstream performance. Indeed, as shown in Table 3, SwAV learns much faster and reaches higher performance in $4\times$ fewer epochs: 72% after 200 epochs (102 hours) while MoCov2 needs 800 epochs to achieve 71.1%. Increasing the resolution and the number of epochs, SwAV reaches 74.3% with a small batch size, a small number of stored features and no momentum encoder. Finally, note that SwAV could be combined with a momentum mechanism and a large queue [20]; we leave these explorations to future work.

# 5 Ablation Study

## 5.1 Clustering-based self-supervised learning

**Improving prior clustering-based approaches.** In this section, we re-implement and improve previously published clustering-based models in order to assess if they can compete with recent contrastive methods such as SimCLR. In particular, we consider two clustering-based models:

| Method | Top-1 | | Δ |
|---|---|---|---|
| | 2x224 | 2x160+4x96 | |
| Supervised | 76.5 | 76.0 | −0.5 |
| *Contrastive-instance approaches* | | | |
| SimCLR | 68.2 | 70.6 | +2.4 |
| *Clustering-based approaches* | | | |
| SeLa-v2 | 67.2 | 71.8 | +4.6 |
| DeepCluster-v2 | 70.2 | 74.3 | +4.1 |
| SwAV | 70.1 | 74.1 | +4.0 |

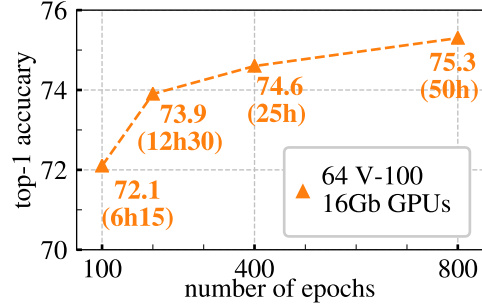

Figure 3: Top-1 accuracy on ImageNet with a linear classifier trained on top of frozen features from a ResNet-50. **(left) Comparison between clustering-based and contrastive instance methods and impact of multi-crop.** Self-supervised methods are trained for 400 epochs and supervised models for 200 epochs. **(right) Performance as a function of epochs.** We compare SwAV models trained with different number of epochs and report their running time based on our implementation.

DeepCluster-v2 and SeLa-v2, which are obtained by applying various training improvements introduced in other self-supervised learning papers to DeepCluster [6] and SeLa [2]. Among these improvements are the use of stronger data augmentation [9], MLP projection head [9], cosine learning rate schedule [37], temperature parameter [49], memory bank [49], multi-clustering [2], etc. Full implementation details can be found in the supplementary material. Besides, we also improve Deep-Cluster model by introducing explicit comparisons to k-means centroids, which increase stability and performance. Indeed, a main issue in DeepCluster is that there is no correspondance between two consecutive cluster assignments. Hence, the final classification layer learned for an assignment becomes irrelevant for the following one and thus needs to be re-initialized from scratch at each epoch. This considerably disrupts the convnet training. In DeepCluster-v2, instead of learning a classification layer predicting the cluster assignments, we perform explicit comparison between features and centroids.

**Comparing clustering with contrastive instance learning.** In Fig. 3 (left), we make a best effort fair comparison between clustering-based and contrastive instance (SimCLR) methods by implementating these methods with the same data augmentation, number of epochs, batch-sizes, etc. In this setting, we observe that SwAV and DeepCluster-v2 outperform SimCLR by 2% without `multi-crop` and by 3.5% with `multi-crop`. This suggests the learning potential of clustering-based methods over instance classification.

**Advantage of SwAV compared to DeepCluster-v2.** In Fig. 3 (left), we observe that SwAV performs on par with DeepCluster-v2. In addition, we train DeepCluster-v2 in SwAV best setting (800 epochs - 8 crops) and obtain 75.2% top-1 accuracy on ImageNet (versus 75.3% for SwAV). However, unlike SwAV, DeepCluster-v2 is not online which makes it impractical for extremely large datasets (§ 5.4). For billion scale trainings for example, a single pass on the dataset is usually performed [20]. DeepCluster-v2 cannot be trained for only one epoch since it works by performing several passes on the dataset to regularly update centroids and cluster assignments for each image.

As a matter of fact, DeepCluster-v2 can be interpreted as a special case of our proposed swapping mechanism: swapping is done across epochs rather than within a batch. Given a crop of an image DeepCluster-v2 predicts the assignment of another crop, which was obtained at the previous epoch. SwAV swaps assignments directly at the batch level and can thus work online.

## 5.2 Applying the multi-crop strategy to different methods

In Fig. 3 (left), we report the impact of applying our `multi-crop` strategy on the performance of a selection of other methods. Details of how we apply `multi-crop` to SimCLR loss can be found in the supplementary material. We see that the `multi-crop` strategy consistently improves the performance for all the considered methods by a significant margin of 2−4% top-1 accuracy. Interestingly, `multi-crop` seems to benefit more clustering-based methods than contrastive methods. We note that `multi-crop` does not improve the supervised model.

| Method | Frozen | Finetuned |
|--------|--------|-----------|
| Random | 15.0 | 76.5 |
| MoCo | - | 77.3* |
| SimCLR | 60.4 | 77.2 |
| SwAV | **66.5** | **77.8** |

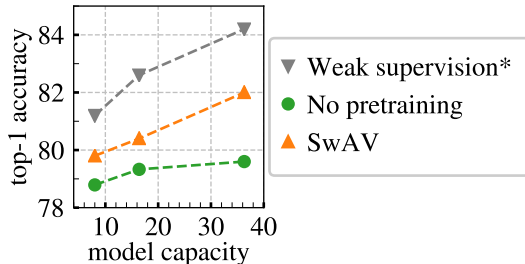

Figure 4: **Pretraining on uncurated data.** Top-1 accuracy on ImageNet for pretrained models on an uncurated set of 1B random Instagram images. **(left)** We compare ResNet-50 pretrained with either SimCLR or SwAV on two downstream tasks: linear classification on frozen features or finetuned features. **(right)** Performance of finetuned models as we increase the capacity of a ResNext following [35]. The capacity is provided in billions of Mult-Add operations.
*: *pretrained on a curated set of* 1*B Instagram images filtered with* 1.5*k hashtags similar to ImageNet classes.*

## 5.3    Impact of longer training

In Fig. 3 (right), we show the impact of the number of training epochs on performance for SwAV with `multi-crop`. We train separate models for 100, 200, 400 and 800 epochs and report the top-1 accuracy on ImageNet using the linear classification evaluation. We train each ResNet-50 on 64 V100 16GB GPUs and a batch size of 4096. While SwAV benefits from longer training, it already achieves strong performance after 100 epochs, *i.e.*, 72.1% in 6h15.

## 5.4    Unsupervised pretraining on a large uncurated dataset

We evaluate SwAV on random, uncurated images that have different properties from ImageNet which allows us to test if our online clustering scheme and multi-crop augmentation work out of the box. In particular, we pretrain SwAV on an uncurated dataset of 1 billion random public non-EU images from Instagram. We test if SwAV can serve as a pretraining method for supervised learning. In Fig. 4 (left), we measure the performance of ResNet-50 models when transferring to ImageNet with frozen or finetuned features. We report the results from He *et al*. [20] but note that their setting is different. They use a curated set of Instagram images, filtered by hashtags similar to ImageNet labels [35]. We compare SwAV with a randomly initialized network and with a network pretrained on the same data using SimCLR. We observe that SwAV maintains a similar gain of 6% over SimCLR as when pretrained on ImageNet (Fig. 2), showing that our improvements do not depend on the data distribution. We also see that pretraining with SwAV on random images significantly improves over training from scratch on ImageNet (+1.3%) [7, 20]. This result is in line with Caron *et al*. [7] and He *et al*. [20]. In Fig. 4 (right), we explore the limits of pretraining as we increase the model capacity. We consider the variants of the ResNeXt architecture [52] as in Mahajan *et al*. [35]. We compare SwAV with supervised models trained from scratch on ImageNet. For all models, SwAV outperforms training from scratch by a significant margin showing that it can take advantage of the increased model capacity. For reference, we also include the results from Mahajan *et al*. [35] obtained with a weakly-supervised model pretrained by predicting hashtags filtered to be similar to ImageNet classes. Interestingly, SwAV performance is strong when compared to this topline despite not using any form of supervision or filtering of the data.

## 6    Discussion

Self-supervised learning is rapidly progressing compared to supervised learning, even surpassing it on transfer learning, even though the current experimental settings are designed for supervised learning. In particular, architectures have been designed for supervised tasks, and it is not clear if the same models would emerge from exploring architectures with no supervision. Several recent works have shown that exploring architectures with search [33] or pruning [8] is possible without supervision, and we plan to evaluate the ability of our method to guide model explorations.

## Broader Impact

This work presents a self-supervised method for learning visual representations. Self-supervised or unsupervised learning allows training models with no annotations, nor metadata. Thus, this work increases the field of possible applications of image features to domains where annotations are hard to collect. For example, removing the need for annotations benefits applications where annotations require expert knowledge, like medical imaging, or are time consuming, like fine-grained classification. This work improves unsupervised feature learning and thus many potential downstream applications that use visual features can benefit from it. We are uncertain of all the possible new applications, but each application has its own merits and societal implications depending on the intentions of the individuals using the technology.

Evaluating visual representations, whether they are supervised or self-supervised, is an open research question. Typically used benchmarks can suffer from dataset or concept bias, and thus may reinforce or guide future research in that direction. To mitigate this, we evaluate our work on multiple different benchmarks and hope that future researchers also take steps in this direction.

**Acknowledgement.** We thank Nicolas Carion, Kaiming He, Herve Jegou, Benjamin Lefaudeux, Thomas Lucas, Francisco Massa, Sergey Zagoruyko, and the rest of Thoth and FAIR teams for their help and fruitful discussions. Julien Mairal was funded by the ERC grant number 714381 (SOLARIS project) and by ANR 3IA MIAI@Grenoble Alpes (ANR-19-P3IA-0003).

## Footnotes

* Univ. Grenoble Alpes, Inria, CNRS, Grenoble INP, LJK, 38000 Grenoble, France

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
