[Supplementary Material]

# Supplementary Material for Unsupervised Learning of Visual Features by Contrasting Cluster Assignments

**Mathilde Caron**[1,2]    **Ishan Misra**[2]    **Julien Mairal**[1]

**Priya Goyal**[2]    **Piotr Bojanowski**[2]    **Armand Joulin**[2]

[1] Inria[*]    [2] Facebook AI Research

## A    Implementation Details

In this section, we provide the details and hyperparameters for SwAV pretraining and transfer learning. Our code is publicly available at https://github.com/facebookresearch/swav.

### A.1    Implementation details of SwAV training

First, we provide a pseudo-code for SwAV training loop using two crops in Pytorch style:

```
# C: prototypes (DxK)
# model:  convnet + projection head
# temp:  temperature

for x in loader: # load a batch x with B samples
  x_t = t(x) # t is a random augmentation
  x_s = s(x) # s is a another random augmentation

  z = model(cat(x_t, x_s)) # embeddings:  2BxD

  scores = mm(z, C) # prototype scores:  2BxK
  scores_t = scores[:B]
  scores_s = scores[B:]

  # compute assignments
  with torch.no_grad():
    q_t = sinkhorn(scores_t)
    q_s = sinkhorn(scores_s)

  # convert scores to probabilities
  p_t = Softmax(scores_t / temp)
  p_s = Softmax(scores_s / temp)

  # swap prediction problem
  loss = - 0.5 * mean(q_t * log(p_s) + q_s * log(p_t))

  # SGD update:  network and prototypes
  loss.backward()
```

```
      update(model.params)
      update(C)

      # normalize prototypes
      with torch.no_grad():
         C = normalize(C, dim=0, p=2)

# Sinkhorn-Knopp
def sinkhorn(scores, eps=0.05, niters=3):
   Q = exp(scores / eps).T
   Q /= sum(Q)
   K, B = Q.shape
   u, r, c = zeros(K), ones(K) / K, ones(B) / B
   for _ in range(niters):
      u = sum(Q, dim=1)
      Q *= (r / u).unsqueeze(1)
      Q *= (c / sum(Q, dim=0)).unsqueeze(0)
   return (Q / sum(Q, dim=0, keepdim=True)).T
```

Most of our training hyperparameters are directly taken from SimCLR work [6]. We train SwAV with stochastic gradient descent using large batches of 4096 different instances. We distribute the batches over 64 V100 16Gb GPUs, resulting in each GPU treating 64 instances. The temperature parameter $\tau$ is set to 0.1 and the Sinkhorn regularization parameter $\varepsilon$ is set to 0.05 for all runs. We use a weight decay of $10^{-6}$, LARS optimizer [27] and a learning rate of 4.8 which is linearly ramped up during the first 10 epochs. After warmup, we use the cosine learning rate decay [20, 22] with a final value of 0.0048. To help the very beginning of the optimization, we freeze the prototypes during the first epoch of training. We synchronize batch-normalization layers across GPUs using the optimized implementation with kernels through CUDA/C-v2 extension from apex[1]. We also use apex library for training with mixed precision [21]. Overall, thanks to these training optimizations (mixed precision, kernel batch-normalization and use of large batches [13]), 100 epochs of training for our best SwAV model take approximately 6 hours (see Table 1). Similarly to previous works [6, 7, 18], we use a projection head on top of the convnet features that consists in a 2-layer multi-layer perceptron (MLP) that projects the convnet output to a 128-D space.

Note that SwAV is more suitable for a multi-node distributed implementation compared to contrastive approaches SimCLR or MoCo. The latter methods require sharing the feature matrix across all GPUs at every batch which might become a bottleneck when distributing across many GPUs. On the contrary, SwAV requires sharing only matrix normalization statistics (sum of rows and columns) during the Sinkhorn algorithm.

## A.2 Data augmentation used in SwAV

We obtain two different views from an image by performing crops of random sizes and aspect ratios. Specifically we use the RandomResizedCrop method from torchvision.transforms module of PyTorch with the following scaling parameters: s=(0.14, 1). Note that we sample crops in a narrower range of scale compared to the default RandomResizedCrop parameters. Then, we resize both full resolution views to $224 \times 224$ pixels, unless specified otherwise (we use $160 \times 160$ resolutions in some of our experiments). Besides, we obtain $V$ additional views by cropping small parts in the image. To do so, we use the following RandomResizedCrop parameters: s=(0.05, 0.14). We resize the resulting crops to $96 \times 96$ resolution. Note that we always deal with resolutions that are divisible by 32 to avoid roundings in the ResNet-50 pooling layers. Finally, we apply random horizontal flips, color distortion and Gaussian blur to each resulting crop, exactly following the SimCLR implementation [6]. An illustration of our multi-crop augmentation strategy can be viewed in Fig. 1.

Figure 1: **Multi-crop**: the image $x_n$ is transformed into $V + 2$ views: two global views and $V$ small resolution zoomed views.

### A.3 Implementation details of linear classification on ImageNet with ResNet-50

We obtain 75.3 top-1 accuracy on ImageNet by training a linear classifier on top of frozen final representations (2048-D) of a ResNet-50 trained with SwAV. This linear layer is trained during 100 epochs, with a learning rate of 0.3 and a weight decay of $10^{-6}$. We use cosine learning rate decay and a batch size of 256. We use standard data augmentations, i.e., cropping of random sizes and aspect ratios (default parameters of `RandomResizedCrop`) and random horizontal flips.

### A.4 Implementation details of semi-supervised learning (finetuning with 1% or 10% labels)

We finetune with either 1% or 10% of ImageNet labeled images a ResNet-50 pretrained with SwAV. We use the 1% and 10% splits specified in the official code release of SimCLR. We mostly follow hyperparameters from PCL [18]: we train during 20 epochs with a batch size of 256, we use distinct learning rates for the convnet weights and the final linear layer, and we decay the learning rates by a factor 0.2 at epochs 12 and 16. We do not apply any weight decay during finetuning. For 1% finetuning, we use a learning rate of 0.02 for the trunk and 5 for the final layer. For 10% finetuning, we use a learning rate of 0.01 for the trunk and 0.2 for the final layer.

### A.5 Implementation details of transfer learning on downstream tasks

**Linear classifiers.** We mostly follow PIRL [22] for training linear models on top of representations given by a ResNet-50 pretrained with SwAV. On VOC07, all images are resized to 256 pixels along the shorter side, before taking a $224 \times 224$ center crop. Then, we train a linear SVM with LIBLINEAR [10] on top of corresponding global average pooled final representations (2048-D). For linear evaluation on other datasets (Places205 and iNat18), we train linear models with stochastic gradient descent using a batch size of 256, a learning rate of 0.01 reduced by a factor of 10 three times (equally spaced intervals), weight decay of 0.0001 and momentum of 0.9. On Places205, we train the linear models for 28 epochs and on iNat18 for 84 epochs. We report the top-1 accuracy computed using the $224 \times 224$ center crop on the validation set.

**Object Detection on VOC07+12.** We use a Faster R-CNN [23] model as implemented in Detectron2 [25] and follow the finetuning protocol from He *et al*. [14] making the following changes to the hyperparameters – our initial learning rate is 0.1 which is warmed with a slope (`WARMUP_FACTOR` flag in Detectron2) of 0.333 for 1000 iterations. Other training hyperparamters are kept exactly the same as in He *et al*. [14], *i.e.*, batchsize of 16 across 8 GPUs, training for 24K iterations on VOC07+12 `trainval` with the learning rate reduced by a factor of 10 after 18K and 22K iterations, using SyncBatchNorm to finetune BatchNorm parameters, and adding an extra BatchNorm layer after the `res5` layer (`Res5ROIHeadsExtraNorm` head in Detectron2). We report results on VOC07 test set averaged over 5 independant runs.

**Object Detection on COCO.** We test the generalization of our ResNet-50 features trained on ImageNet with SwAV by transferring them to object detection on COCO dataset [19] with DETR framework [4]. DETR is a recent object detection framework that relies on a transformer encoder-decoder architecture. It reaches competitive performance with Faster R-CNN while being conceptually

simpler and trainable end-to-end. Interestingly, unlike other frameworks [15], current results with DETR have shown that using a pretrained backbone is crucial to obtain good results compared to training from scratch. Therefore, we investigate if we can boost DETR performance by using features pretrained on ImageNet with SwAV instead of standard supervised features. We also evaluate features from MoCov2 [7] pretraining. We train DETR during 300 epochs with AdamW, we use a learning rate of $10^{-4}$ for the transformer and apply a weight decay of $10^{-4}$. We select for each method the best learning rate for the backbone among the following three values: $10^{-5}$, $5 \times 10^{-5}$ and $10^{-4}$. We decay the learning rates by a factor $0.1$ after $200$ epochs.

## A.6 Implementation details of training with small batches of 256 images

We start using a queue composed of the feature representations from previous batches after $15$ epochs of training. Indeed, we find that using the queue before $15$ epochs disturbs the convergence of the model since the network is changing a lot from an iteration to another during the first epochs. We simulate large batches of size $4096$ by storing the last $15$ batches, that is $3,840$ vectors of dimension $128$. We use a weight decay of $10^{-6}$, LARS optimizer [27] and a learning rate of $0.6$. We use the cosine learning rate decay [20] with a final value of $0.0006$.

## A.7 Implementation details of ablation studies

In our ablation studies (results in Table 5 of the main paper for example), we choose to follow closely the data augmentation used in concurrent work SimCLR. This allows a fair comparison and importantly, isolates the effect of our contributions. In practice, it means that we use the default parameters of the random crop method (`RandomResizedCrop`), `s=(0.08, 1)` instead of `s=(0.14, 1)`, when sampling the two large resolution views.

## A.8 SimCLR loss with multi-crop augmentation

When implementing SimCLR with `multi-crop` augmentation, we have to deal with several positive pairs formed by an anchor features and the different crops coming from the same instance. We denote by $B$ the total number of unique dataset instances in the batch and by $M$ the number of crops per instance. For example, in the case of 2x160+4x96 crops, we have $M = 6$ crops per instance. We call $N = B \times M$ the effective total number of crops in the batch. Overall, we minimize the following loss

$$\mathcal{L} = -\frac{1}{N}\frac{1}{M-1}\sum_{i=1}^{N}\sum_{v^+\in\{v_i^+\}}\log\frac{\exp z_i^T v^+/\tau}{\exp z_i^T v^+/\tau + \sum_{v^-\in\{v_i^-\}}\exp z_i^T v^-/\tau}. \tag{1}$$

For a feature representation $z_i$, the corresponding set of positive examples $\{v_i^+\}$ is formed by the representations of the other crops from the same instance. The set of negatives $\{v_i^-\}$ is formed by the representations of all crops in the same batch except ones coming from the same instance as $x_i$. Note that this extension of SimCLR loss with several pairs of positive is similar to the one used in Khosla *et al.* [17].

## B  Additional Results

### B.1  Running times

In Table 1, we report compute and GPU memory requirements based on our implementation for different settings. As described in § A.1, we train each method on $64$ V100 16GB GPUs, with a batch size of $4096$, using mixed precision and `apex` optimized version of synchronized batch-normalization layers. We report results with ResNet-50 for all methods. In Fig. 2, we report SwAV performance for different training lengths measured in hours based on our implementation. We observe that after only 6 hours of training, SwAV outperforms SimCLR trained for 1000 epochs (40 hours based on our implementation) by a large margin. If we train SwAV for longer, we see that the performance gap between the two methods increases even more.

Table 1: **Computational cost.** We report time and GPU memory requirements based on our implementation for different models trained during 100 epochs.

| Method | `multi-crop` | time / 100 epochs | peak memory / GPU |
|---|---|---|---|
| SimCLR | $2 \times 224$ | 4h00 | 8.6G |
| SwAV | $2 \times 224$ | 4h09 | 8.6G |
| SwAV | $2 \times 160 + 4 \times 96$ | 4h50 | 8.5G |
| SwAV | $2 \times 224 + 6 \times 96$ | 6h15 | 12.8G |

Figure 2: **Influence of longer training.** Top-1 ImageNet accuracy for linear models trained on frozen features. We report SwAV performance for different training lengths measured in hours based on our implementation. We train each ResNet-50 models on 64 V100 16GB GPUs with a batch size of 4096 (see § A.1 for implementation details).

## B.2 Larger architectures

In Table 2, we show results when training SwAV on large architectures. We observe that SwAV benefits from training on large architectures and plan to explore in this direction to furthermore boost self-supervised methods.

Table 2: **Large architectures.** Top-1 accuracy for linear models trained on frozen features from different self-supervised methods on large architectures.

| Method | Arch. | Param. | Top1 |
|---|---|---|---|
| Supervised | EffNet-B7 | 66 | 84.4 |
| Rotation [12] | RevNet50-4w | 86 | 55.4 |
| BigBiGAN [9] | RevNet50-4w | 86 | 61.3 |
| AMDIM [2] | Custom-RN | 626 | 68.1 |
| CMC [24] | R50-w2 | 188 | 68.4 |
| MoCo [14] | R50-w4 | 375 | 68.6 |
| CPC v2 [16] | R161 | 305 | 71.5 |
| SimCLR [6] | R50-w4 | 375 | 76.8 |
| SwAV | R50-w2 | 188 | **77.3** |
| SwAV | R50-w4 | 375 | **77.9** |
| SwAV | R50-w5 | 586 | **78.5** |

**Implementation details for SwAV R50-w2.** The model is trained for 400 epochs on 128 GPUS (batch size 4096). We train the model with 2x224+4x96 (total of 6 crops). All other hyperparameters are the same as the ones described in appendix A.1.

**Implementation details for SwAV R50-w4.** The model is trained for 400 epochs on 64 GPUS (batch size 2560) with a queue of 2560 samples starting from the beginning of training. We train the model with 2x224+4x96 (total of 6 crops). All other hyperparameters are the same as the ones described in appendix A.1.

**Implementation details for SwAV R50-w5.** The model is trained for 400 epochs on 128 GPUS (batch size 1536) with a queue of 1536 samples starting from the beginning of training. We train the model with 2x224+4x96 (total of 6 crops). All other hyperparameters are the same as the ones described in appendix A.1.

## B.3 Transferring unsupervised features to downstream tasks

In Table 3, we expand results from the main paper by providing numbers from previously and concurrently published self-supervised methods. In the left panel of Table 3, we show performance after training a linear classifier on top of frozen representations on different datasets while on the right panel we evaluate the features by finetuning a ResNet-50 on object detection with Faster R-CNN [23] and DETR [4]. Overall, we observe on Table 3 that SwAV is the first self-supervised method to outperform ImageNet supervised backbone on all the considered transfer tasks and datasets. Other self-supervised learners are capable of surpassing the supervised counterpart but only for one type of transfer (object detection with finetuning for MoCo/PIRL for example). We will release this model so other researchers might also benefit by replacing the ImageNet supervised network with our model.

Table 3: **Transfer learning on downstream tasks.** Comparison between features from ResNet-50 trained on ImageNet with SwAV or supervised learning. We also report numbers from other self-supervised methods ([†] for numbers from other methods run by us). We consider two settings. (1) Linear classification on top of frozen features. We report top-1 accuracy on Places205 and iNat18 datasets and mAP on VOC07. (2) Object detection with finetuned features on VOC07+12 `trainval` using Faster R-CNN [23] and on COCO [19] using DETR [4]. In this table, we report the most standard detection metrics for these datasets: $AP_{50}$ on VOC07+12 and AP on COCO.

| | Linear Classification | | | Object Detection | |
|---|---|---|---|---|---|
| | Places205 | VOC07 | iNat18 | VOC07+12 (Faster R-CNN) | COCO (DETR) |
| Supervised | 53.2 | 87.5 | 46.7 | 81.3 | 40.8 |
| RotNet [11] | 45.0 | 64.6 | - | - | - |
| NPID++ [22] | 46.4 | 76.6 | 32.4 | 79.1 | - |
| MoCo [14] | 46.9[†] | 79.8[†] | 31.5[†] | 81.5 | - |
| PIRL [22] | 49.8 | 81.1 | 34.1 | 80.7 | - |
| PCL [18] | 49.8 | 84.0 | - | - | - |
| BoWNet [11] | 51.1 | 79.3 | - | 81.3 | - |
| SimCLR [6] | 53.3[†] | 86.4[†] | 36.2[†] | - | - |
| MoCov2 [14] | 52.9[†] | 87.1[†] | 38.9[†] | 82.5 | 42.0[†] |
| SwAV | **56.7** | **88.9** | **48.6** | **82.6** | **42.1** |

## B.4 More detection metrics for object detection

In Table 4 and Table 5, we evaluate the features by finetuning a ResNet-50 on object detection with Faster R-CNN [23] and DETR [4] and report more detection metrics compared to Table 3. We observe in Table 4 and in Table 5 that SwAV outperforms the ImageNet supervised pretrained model on all the detection evaluation metrics. Note that MoCov2 backbone performs particularly well on the object detection benchmark, and even outperform SwAV features for some detection metrics. However, as shown in Table 3, this backbone is not competitive with the supervised features when evaluating on classification tasks without finetuning.

## B.5 Low-Shot learning on ImageNet for SwAV pretrained on Instagram data

We now test whether SwAV pretrained on Instagram data can serve as a pretraining method for low-shot learning on ImageNet. We report in Table 5 results when finetuning Instagram SwAV features with only few labels per ImageNet category. We observe that using pretrained features from Instagram improves considerably the performance compared to training from scratch.

## B.6 Image classification with KNN classifiers on ImageNet

Following previous work protocols [26, 28], we evaluate the quality of our unsupervised features with K-nearest neighbor (KNN) classifiers on ImageNet. We get features from the computed network outputs for center crops of training and test images. We report results with 20 and 200 NN in Table 7. We outperform the current state-of-the-art of this evaluation. Interestingly we also observe that using fewer NN actually boosts the performance of our model.

Table 4: **More detection metrics for object detection on VOC07+12 with finetuned features using Faster R-CNN [23].**

| Method | $AP^{all}$ | $AP^{50}$ | $AP^{75}$ |
|---|---|---|---|
| Supervised | 53.5 | 81.3 | 58.8 |
| Random | 28.1 | 52.5 | 26.2 |
| NPID++ [22] | 52.3 | 79.1 | 56.9 |
| PIRL [22] | 54.0 | 80.7 | 59.7 |
| BoWNet [11] | 55.8 | 81.3 | 61.1 |
| MoCov1 [14] | 55.9 | 81.5 | 62.6 |
| MoCov2 [7] | **57.4** | 82.5 | **64.0** |
| SwAV | 56.1 | **82.6** | 62.7 |

Table 5: **More detection metrics for object detection on COCO with finetuned features using DETR [4].**

| Method | AP | $AP_{50}$ | $AP_{75}$ | $AP_S$ | $AP_M$ | $AP_L$ |
|---|---|---|---|---|---|---|
| ImageNet labels | 40.8 | 61.2 | 42.9 | 20.1 | 44.5 | 60.3 |
| MoCo-v2 | 42.0 | 62.7 | 44.4 | **20.8** | 45.6 | **60.9** |
| SwAV | **42.1** | **63.1** | **44.5** | 19.7 | **46.3** | **60.9** |

## C  Ablation Studies on Clustering

### C.1  Number of prototypes

In Table 8, we evaluate the influence of the number of prototypes used in SwAV. We train ResNet-50 with SwAV for 400 epochs with $2 \times 160 + 4 \times 96$ crops (ablation study setting) and evaluate the performance by training a linear classifier on top of frozen final representations. We observe in Table 8 that varying the number of prototypes by an order of magnitude (3k-100k) does not affect much the performance (at most $0.3$ on ImageNet). This suggests that the number of prototypes has little influence as long as there are "enough". Throughout the paper, we train SwAV with 3000 prototypes. We find that using more prototypes increases the computational time both in the Sinkhorn algorithm and during back-propagation for an overall negligible gain in performance.

### C.2  Learning the prototypes

We investigate the impact of learning the prototypes compared to using fixed random prototypes. Assigning features to fixed random targets has been explored in NAT [3]. However, unlike SwAV, NAT uses a target per instance in the dataset, the assignment is hard and performed with Hungarian algorithm. In Table 9 (left), we observe that learning prototypes improves SwAV from 73.1 to 73.9 which shows the effect of adapting the prototypes to the dataset distribution.

Overall, these results suggest that our framework learns from a different signal from "offline" approaches that attribute a pseudo-label to each instance while considering the full dataset and then predict these labels (like DeepCluster [5] for example). Indeed, the prototypes in SwAV are not strongly encouraged to be categorical and random fixed prototypes work almost as well. Rather, they help contrasting different image views without relying on pairwise comparison with many negatives samples. This might explain why the number of prototypes does not impact the performance significantly.

### C.3  Hard versus soft assignments

In Table 9 (right), we evaluate the impact of using hard assignment instead of the default soft assignment in SwAV. We train the models during 400 epochs with $2 \times 160 + 4 \times 96$ crops (ablation study setting) and evaluate the performance by training a linear classifier on top of frozen final representations. We also report the training losses in Fig. 3. We observe that using the hard

Table 6: **Low-shot learning on ImageNet.** Top-1 and top-5 accuracies when training with 13 or 128 examples per category.

| # examples per class | 13 | | 128 | |
|---|---|---|---|---|
| | top1 | top5 | top1 | top5 |
| No pretraining | 25.4 | 48.4 | 56.4 | 80.4 |
| SwAV IG-1B | **38.2** | **67.1** | **64.7** | **87.2** |

Table 7: **KNN classifiers on ImageNet.** We report top-1 accuracy with 20 and 200 nearest neighbors.

| Method | 20-NN | 200-NN |
|---|---|---|
| NPID [26] | - | 46.5 |
| LA [28] | - | 49.4 |
| PCL [18] | 54.5 | - |
| SwAV | **59.2** | **55.8** |

assignments performs worse than using the soft assignments. An explanation is that the rounding needed to obtain discrete codes is a more aggressive optimization step than gradient updates. While it makes the model converge rapidly (see Fig. 3), it leads to a worse solution.

### C.4 Impact of the number of iterations in Sinkhorn algorithm

In Table 10, we investigate the impact of the number of normalization steps performed during Sinkhorn-Knopp algorithm [8] on the performance of SwAV. We observe that using only 3 iterations is enough for the model to converge. When performing less iterations, the loss fails to converge. We observe that using more iterations slightly alters the transfer performance of the model. We conjecture that it is for the same reason that rounding codes to discrete values deteriorate the quality of our model by converging too rapidly.

### D Details on Clustering-Based methods: DeepCluster-v2 and SeLa-v2

In this section, we provide details on our improved implementation of clustering-based approaches DeepCluster-v2 and SeLa-v2 compared to their corresponding original publications [1, 5]. These two methods follow the same pipeline: they alternate between pseudo-labels generation ("assignment phase") and training the network with a classification loss supervised by these pseudo-labels ("training phase").

Figure 3: **Hard versus soft assignments.** We report the training loss for SwAV models trained with either soft or hard assignments. The models are trained during $400$ epochs with $2 \times 160 + 4 \times 96$ crops.

Table 8: **Impact of number of prototypes.** Top-1 ImageNet accuracy for linear models trained on frozen features.

| Number of prototypes | 300 | 1000 | 3000 | 10000 | 30000 | 100000 |
|---|---|---|---|---|---|---|
| Top-1 | 72.8 | 73.6 | 73.9 | 74.1 | 73.8 | 73.8 |

Table 9: **Ablation studies on clustering.** Top-1 ImageNet accuracy for linear models trained on frozen features. **(left)** Impact of learning the prototypes. **(right)** Hard versus soft assignments.

| Prototypes | Learned | Fixed | | Assignment | Soft | Hard |
|---|---|---|---|---|---|---|
| Top-1 | 73.9 | 73.1 | | Top-1 | 73.9 | 73.3 |

## D.1 Training phase

During the training phase, both methods minimize the multinomial logistic loss of the pseudo-labels $\mathbf{q}$ classification problem:

$$\ell(\mathbf{z}, \mathbf{c}, \mathbf{q}) = -\sum_k \mathbf{q}^{(k)} \log \mathbf{p}^{(k)}, \quad \text{where} \quad \mathbf{p}^{(k)} = \frac{\exp\left(\frac{1}{\tau}\mathbf{z}^\top \mathbf{c}_k\right)}{\sum_{k'} \exp\left(\frac{1}{\tau}\mathbf{z}^\top \mathbf{c}_{k'}\right)}. \tag{2}$$

The pseudo-labels are kept fixed during training and updated for the entire dataset once per epoch during the assignment phase.

**Training phase in DeepCluster-v2.** In the original DeepCluster work, both the classification head $\mathbf{c}$ and the convnet weights are trained to classify the images into their corresponding pseudo-label between two assignments. Intuitively, this classification head is optimized to represent prototypes for the different pseudo-classes. However, since there is no mapping between two consecutive assignments: the classification head learned during an assignment becomes irrelevant for the following one. Thus, this classification head needs to be re-set at each new assignment which considerably disrupts the convnet training. For this reason, we propose to simply use for classification head $\mathbf{c}$ the centroids given by k-means clustering (Eq. 5). Overall, during training, DeepCluster-v2 optimizes the following problem with mini-batch SGD:

$$\min_{\mathbf{z}} \ell(\mathbf{z}, \mathbf{c}, \mathbf{q}). \tag{3}$$

**Training phase in SeLa-v2.** In SeLa work, the prototypes $\mathbf{c}$ are learned with stochastic gradient descend during the training phase. Overall, during training, SeLa-v2 optimizes the following problem:

$$\min_{\mathbf{z}, \mathbf{c}} \ell(\mathbf{z}, \mathbf{c}, \mathbf{q}). \tag{4}$$

## D.2 Assignment phase

The purpose of the assignment phase is to provide assignments $\mathbf{q}$ for each instance of the dataset. For both methods, this implies having access to feature representations $\mathbf{z}$ for the entire dataset. Both original works [1, 5] perform regularly a pass forward on the whole dataset to get these features. Using the original implementation, if assignments are updated at each epoch, then the assignment phase represents one third of the total training time. Therefore, in order to speed up training, we choose to use the features computed during the previous epoch instead of dedicating pass forwards to the assignments. This is similar to the memory bank introduced by Wu *et al.* [26], without momentum.

Table 10: **Impact of the number of iterations in Sinkhorn algorithm.** Top-1 ImageNet accuracy for linear models trained on frozen features.

| Sinkhorn iterations | 1 | 3 | 10 | 30 |
|---|---|---|---|---|
| Top-1 | *fail* | 73.9 | 73.8 | 73.7 |

**Assignment phase in DeepCluster-v2.** DeepCluster-v2 uses spherical k-means to get pseudo-labels. In particular, pseudo-labels $\mathbf{q}$ are obtained by minimizing the following problem:

$$\min_{\mathbf{C} \in \mathbb{R}^{d \times K}} \frac{1}{N} \sum_{n=1}^{N} \min_{\mathbf{q}} -\mathbf{z}_n^\top \mathbf{C} \mathbf{q}, \tag{5}$$

where $\mathbf{z}_n$ and the columns of $\mathbf{C}$ are normalized. The original work DeepCluster uses tricks such as cluster re-assignments and balanced batch sampling to avoid trivial solutions but we found these unnecessary, and did not observe collapsing during our trainings. As noted by Asano *et al.*, this is due to the fact that assignment and training are well separated phases.

**Assignment phase in SeLa-v2.** Unlike DeepCluster, SeLa uses the same loss during training and assignment phases. In particular, we use Sinkhorn-Knopp algorithm to optimize the following assignment problem (see details and derivations in the original SeLa paper [1]):

$$\min_{\mathbf{q}} \ell(\mathbf{z}, \mathbf{c}, \mathbf{q}). \tag{6}$$

**Implementation details** We use the same hyperparameters as SwAV to train SeLa-v2 and DeepCluster-v2: these are described in § A. Asano *et al*. [1] have shown that multi-clustering boosts performance of clustering-based approaches, and so we use 3 sets of 3000 prototypes $\mathbf{c}$ when training SeLa-v2 and DeepCluster-v2. Note that unlike online methods (like SwAV, SimCLR and MoCo), the clustering approaches SeLa-v2 and DeepCluster-v2 can be implemented with only a single crop per image per batch. The major limitation of SeLa-v2 and DeepCluster-v2 is that these methods are not online and therefore scaling them to very large scale dataset is not posible without major adjustments.

## Footnotes

[1]github.com/NVIDIA/apex