[Reviews · NeurIPS 2020]

Review 1

Summary and Contributions: Unsupervised learning of image representations that rival or surpass supervised representations is an important topic that has become quite successful of late with CPC, CMC, MoCo and SimCLR. On the other hand, clustering based approaches such as DeepCluster haven't been pushed to a similar extent with more recent engineering practices such as data augmentations, large batch sizes, projection heads, compute scale. This paper is an attempt to push a clustering based self-supervision method to achieve SoTA numbers on linear classification with a ResNet-50 as well as some good numbers on object detection, semi-supervised learning. They propose to enforce consistencies between augmented views of the same data point by ensuring their cluster assignments should be the same, and thus can be swapped as labels in a DeepCluster like training. An extra change on top of DeepCluster is to address the problem of how to efficiently perform online clustering and distill the result into a model. The use of sinkhorn iteration without requiring a global dataset clustering and enforcing the results of it through a swapped data augmentation mechanism is an elegant and simple idea that seems to work well in practice.

Strengths: The paper is simple and empirically well done. The results presented are clearly good and useful to know for the community. It improves on top of the accuracy presented in MoCov2 on ResNet-50 and is competitive with the recently released BYOL (a parallel submission). The idea of trying to use multiple views (more than 2) without having a combinatorial explosion of the affinity matrix is sensible. Similarly, using a local batch size based clustering with swapped augmentation is also a practically good idea for making clustering based self-supervision work online (which I think is the main contribution of this paper though it is not highlighting it enough according to me).

Weaknesses: The paper has many weak points unfortunately. They are presented below as separate categories. Intro/Motivation: The paper focuses too much on “not using momentum encoder”, “not using memory bank”. All these are largely irrelevant points. Firstly, until one shows one gets no benefit from momentum encoder, it is best not to claim that “not having momentum” is a contribution / a positive aspect of the model. Every paper in the topic has observed gain from momentum encoder, ex MoCo, SimCLR-v2, BYOL. There’s no benefit from not using it. It is better to show gains from the momentum mechanism for proposed method too or not focus on that point. Secondly, using a memory bank is also not a bad idea. When someone wants to get the model to work with a small batch size and 4 GPUs, MoCo with a memory bank seems like the go-to solution at the moment. The motivation to use it is completely different. The training time for 200 epochs with bs=256 is 7.3 days for SwAV (proposed method), while for MoCo, it’s just 53 hours (slightly over 2 days). That’s a huge significant boost in training time / wall clock efficiency. By not using momentum encoder or memory bank, one loses performance and wall clock efficiency. The authors are making up for it through multi-crop augmentation which adds more inefficiency by requiring more forward passes. So, the introduction is quite flawed imo because it claims removal of well established and useful pieces in the literature as positive aspects while in reality the method is less efficient if you care about widespread adoption, fast training and a general method that can just work on any image dataset with minimal changes. A pedantic but important comment: The paper seems to create artificial problems that don’t exist and highlight they are solving them. Example, in semi-supervised learning, they separately highlight methods that use RandAugment, or in Instagram results, they highlight methods that use curated Instagram. These are moot points. The authors have access to these modules and datasets (they are open sourced or available), and could run their method in those settings and demonstrate gains on them. Further, the MoCo result on Instagram is not updated to use the multi-crop augmentation. Currently, the results make it look like the proposed method can beat MoCo even though it was uncurated data. These are not empirically well founded claims and also not interesting given a curated version of the dataset already exists. Showing results on both settings for both models is more useful. Curating datasets is also not a super hard problem with current engineering infrastructure. Implementation: I went through the public released code and noticed that the authors have hacked the area_range for the random_crop data augmentation to (0.15, 1) and (0.05, 0.15) for the 224 and 96 crops. This is quite a non-standard choice especially given that SimCLR, MoCo adopt the (0.08, 1) for the 224 crops and so does supervised ResNet. I would like to know how much this impacted the results. Even if it doesn't matter or change conclusions, I would like to know why there has been a change from the standard hparams every other paper uses. DeepCluster v2: It seems that in Figure 3, Deep Cluster v2 is working better than the proposed method SwAV. I also noticed the 75.3 number for SwAV uses 6 96**2 crops and 800 epochs of training, while 74.3 from DeepCluster v2 uses 4 crops and 400 epochs of training and 160**2 crops. I would like to know if there's any benefit from SwAV (swapping mechanism) at all or are the results merely an outcome of multi-crop augmentation, using new hparams for the random crops, using more crops than baselines, and training longer. The authors must present a stronger DeepCluster-v2 baseline trained for 800 epochs, using 6 96**2 crops and 2 224**2 crops and still show SwAV is better. If not, the swapping mechanism does *not* seem important, and the new numbers are an outcome of new data augmentation and improved training (which I really respect as a practitioner and think is important for the field). However, the paper tries to sell the new model/idea more than the tricks involved, and that seems to be a flaky presentation unless the ablation is addressed. Efficiency comparison to DeepCluster-v2: Reported training time for an epoch is roughly the same for both DeepCluster-v2 and SwAV. I would imagine SwAV should be more effective because of the removal of the need to do global clustering. I would like to see the overall training time wall-clock of SwAV and DeepCluster-v2 to see how much overhead the global K-means adds. The authors should focus on this imo since this maybe a nice engineering solution for large scale datasets like Instagram on which something like DeepCluster maybe hard to scale. If the authors present results for overall wall clock time for both DeepCluster-v2 (with a well optimized distributed K-means) and SwAV for both ImageNet and Instagram, that would be illuminating and useful even though epoch time maybe similar. Rather than presenting SwAV as a novelty, it would be better to present it as a fix to making DeepCluster scalable (if indeed the wall clock numbers reflect that). Comparison to SimCLR / MoCo and using data augmentation for linear probe: To my knowledge, SimCLR and MoCo don’t use data augmentation for linear probe training. This is clearly mentioned in the SimCLR paper and they also point out 70+ numbers for the linear probe in case it is trained with augmentations. On the other hand, SwAV seems to be using the multi-crop augmentation even for the linear probe benchmark. This may give it extra advantage over baselines. Would be necessary to report both w and w/o this augmentation similar to SimCLR. Object Detection Results: There is a high variance in the PASCAL benchmark, so the authors should report the results of three or five runs with standard deviation. I believe the MoCo benchmark numbers are averaged results and since the reported number (82.6) is so close to 82.5 from MoCo, it would be good to know if there’s an actual improvement or not. Is there a reason there is no Mask/Faster-RCNN results for COCO similar to MoCo (with FPN and c4)? That would make for a more direct comparison compared to reporting on DeTR (which is also interesting finding but no other self-supervision benchmark on it). Poor Scalability to larger models and potentially overfitting results to R50 (lack of generality): One peculiar but important weakness of this paper is the method doesnt seem to scale well to larger models. Example, the results in BYOL from R50-x4 is better than the result of SwAV with R50-x5 (even with fewer parameters). Same for R50-x2 (76.7 vs 77.4). So it is pretty clear from these results that a lot of the multi-crop tricks are just too specific and overfitted to the ResNet-50 results. R50 is definitely one of the most common architectures and showing good results on it is helpful, but focusing too much on it with a lot of extra hacks just to push up numbers makes the technique less general and useful. Multi-crop does not seem like a good method especially when you move to larger models because forward passes will also become expensive in such a setting. The paper does not show any wall clock time comparisons between SimCLR and their model for larger models. I suspect the proposed method will be much less efficient there. Finally, it is also hard to scale / use such a technique in a general-purpose setting where you can just throw a new dataset or a new architecture or image sizes and it works out of the box. MoCo/SimCLR/BYOL have such properties. SwAV doesn’t because one needs to figure out what the multi-crop aspect ratios are, incur expenses when moving to larger models (both in terms of forward/backward props). POST REBUTTAL: I am satisfied with the author response to address my concerns. my score has been updated.

Correctness: There are some issues (pointed out in the Weaknesses).

Clarity: Largely yes.

Relation to Prior Work: Yes.

Reproducibility: No

Additional Feedback: I am happy to revise scores if the authors address my empirical concerns and the presentation. Re Reproducibility: The paper itself doesn't have all details but the code release does.


Review 2

Summary and Contributions: The paper proposes a clustering-based unsupervised learning method for visual pretraining. The cluster assignment of an image view is predicted by the feature representation from another view. Unlike the recent contrastive learning methods, the swapped prediction scheme does not rely on negative pairs of samples to be compared. Empirical results show strong classification performance on linear probe protocol and semi-supervised settings.

Strengths: * The paper studies a timely research problem, and advances it to the state-of-the-arts. The gap between supervised and unsupervised learning is further reduced. * A swapped prediction formulation, where two views of an image are processed asymmetrically for unsupervised learning. * A multi-crop data augmentation method which significantly improves the performance up to 4 percent with more views per image at a mild computation cost.

Weaknesses: * While the paper presents a comprehensive set of experiments, it is still unclear to me what the baseline method to be compared with, i.e., in a fair setting, without the multi-crop augmentation, how does Swav perform over moco or simclr? * In Figure 3 (left), it seems that the proposed method underperforms deepcluster-v2. I think it worth more introduction and details about deepcluster-v2. How it is improved over the initial version? What is the advantage of Swav given deepcluster-v2 is stronger? * In the supplementary material, the transfer performance to object detection is quite limited, especially on APs with large IoU threshold.

Correctness: Overall, I hesitate to classify this work as a clustering-based approach. I believe the term "clustering" should be re-evaluated to describe this work. Clustering usually refer to assign pseudo-labels to samples, and usually the size of a cluster is larger than one, where multiple samples are tightly grouped together. However in this work, the so-called clustering process essentially spreads 3k instances into 3k prototypes. To avoid trivial solution, Sinkhorn-Knopp is used to renomalize the cluster assignment to achieve the equipartition constraint.

Clarity: The paper is easy to follow and the graphics are good the understand.

Relation to Prior Work: Yes, the paper clearly discusses the relation to prior work, especially on DeepCluster and SeLa.

Reproducibility: Yes

Additional Feedback: Overall, I am even more surprised to find that deepclusterv2 performs so well. The authors did not address my concern over the claim of "clustering based approach" for Swav. Hope that more details/justifications can be provided to support this in the camera ready.


Review 3

Summary and Contributions: This paper proposed a self-supervised learning method that applies online clustering to contrastive self-supervised learning. This paper learns the clusters online and forces different views from a image predict each other’s cluster codes. The proposed method leads to good performance even with a small mini-batch during pre-training.

Strengths: + Solid results on ImageNet linear classification task, as well as other downstream tasks. This paper also shows that in a transfer learning linear classification setup, self-supervised embeddings outperforms supervised embeddings. + Good performance with small mini-batch size. By using online clustering, this paper is able to reduce the mini-batch size to 256 and still achieve comparable performance comparing with MoCov2 with much smaller stored features. + A simple yet effective multi-crop trick that improves multiple self-supervised learning techniques.

Weaknesses: Linear classification on ImageNet is a legacy evaluation protocol that evaluates a self-supervised model (without labels), on 100% supervised labels. Although the natural question is, why don’t we use 100% labels at the first place, linear classification is still a good proxy to rank self-supervised methods on ImageNet. However, it is not optimal to evaluate the self-supervised models on a downstream task with linear classification again, as finetuning a model probably leads to better performance and suites the real-world applications [1]. For example, on Places205 dataset, a training from scratch baseline using VGG achieves 58.9% top-1 accuracy [2], and ResNet50 achieves 61.6% top-1 accuracy [3]. Both methods perform better than the reported 56.7% linear transfer learning setup in this paper. It would be nice to show finetune results on Places205, VOC07 and iNat18, to demonstrate the effectiveness & benefit of the self-supervised models in the practical setup. [1] A Large-scale Study of Representation Learning with the Visual Task Adaptation Benchmark, arXiv:1910.04867. [2] Places: An Image Database for Deep Scene Understanding, arXiv:1610.02055. [3] Revisiting Self-Supervised Visual Representation Learning, CVPR 2019.

Correctness: The method and the claims are well verified by the empirical evaluations.

Clarity: This paper is well written, with enough illustrations, technical details and ablation test results.

Relation to Prior Work: This paper well discussed the differences and contributions to the previous contributions.

Reproducibility: Yes

Additional Feedback: - Missing citation [1]: this paper follows the semi-supervised learning protocol, and uses the Supervised baseline numbers (in Table 1) reported by [1] in section 4.1. - It would be nice to include more results on semi-supervised learning experiments with wider architectures in Table 1. For example, CPCv2 [2] and SimCLR [3] reported 91.2% and 92.6% top-5 accuracy with 10% labels, S4L [1] reported 91.2% top-5 accuracy with 10% labels. It would add a lot of value to the community (for future comparison), by reporting wider ResNet results on 1% and 10% benchmarks. [1] S4L: Self-Supervised Semi-Supervised Learning, ICCV 2019. [2] Data-Efficient Image Recognition with Contrastive Predictive Coding, arXiv:1905.09272. [3] A Simple Framework for Contrastive Learning of Visual Representations, ICML 2020. ------------------ I would like to thank the authors for their additional experiments on finetune and wider architectures. I hope in the camera ready version, the authors could report both top-5 accuracy and top-1 accuracy, as top-5 accuracy has been used as the default metric in 1%/10% finetune according to [4]. [4] https://paperswithcode.com/sota/semi-supervised-image-classification-on-2


Review 4

Summary and Contributions: Submission 1375 is concerned with the problem of unsupervised/self-supervised learning of visual representations. To address this problem, several formulations have been proposed. One involves combining clustering and representation learning, where the clustering step provides pseudo-labels for the representation-learning step. A disadvantage of such approaches is that the clustering and representation-learning steps are generally conducted alternatively, thus slowing-down convergence. Another approach based on the contrastive loss involves comparing pairs of image representations to push-away representations from different images while pulling together transformed versions of the same image. This works well but required very large batches or with specialized training techniques (cf the momentum encoder of Moco). In this work, it is proposed to combine the best of both worlds. The authors draw on the idea of contrastive losses. But instead of comparing directly the representations derived from the encoder, they do so indirectly by comparing the cluster assignments of the encoded features. To do so, the authors heavily rely on the framework of Asano et al. [2]: the compatibility between two cluster assignments is measured as the cross-entropy between soft probabilistic assignments of representations to clusters. This leads to an optimal transport problem that can be solved efficiently using the Sinkhorn-Knopp algorithm. One of the contributions of this work is to adapt the approach of [2] to the online setting. Another (minor) contribution of this paper is to propose a new multi-crop data augmentation strategy. Classification results on the ImageNet dataset and on downstream tasks (feature transfer) validate the proposed approach.

Strengths: The paper is quite clear and reads easily. The combination of contrastive loss and clustering approach is novel, to the best of my knowledge. The reported results are excellent, even when compared to recent results (e.g. SimCLR). The authors especially did an effort to integrate in their implementation recent improvements used in SimCLR,. They also did an effort to port these improvements on two other methods (DeepCluster [6] and SelLa [2]) for fair comparison, which is great. They also tested the effects of the multi-crop data-augmentation on these methods, which is great too.

Weaknesses: I do not see any strong weakness with this work. While it is not radically novel (it builds heavily on previous works and especially [2]), I think it contains sufficient novelty to warrant publication. IMHO, the main issue (which is not restricted to this paper) is that fair comparison with previous works is challenging. Indeed, it can be difficult to understand whether the improved results are due to the proposed idea or to unrelated tweaks. In this work, the authors did an effort to reimplement many of the improvements used in SimCLR on DeepCluster [6] and SeLa [2] – see above. They also conducted experiments with and without multi-crop on several methods (including their own). While these results can be found in different tables, I believe it would be much clearer if they were combined in a single table.

Correctness: The claims and method appear to be correct.

Clarity: The paper is generally clear and reads easily. I would just recommend merging results with/without the SimCLR improvements and with/without multi-crop for all methods (proposed, DeepCluster, SeLa) in a single table. And to have in the same table the previous state-of-the-art.

Relation to Prior Work: Good relation to prior work, including many recent references.

Reproducibility: Yes

Additional Feedback: No further feedback.

[Author Response · NeurIPS 2020]

We thank the reviewers for their detailed and thoughtful comments. Before addressing their remarks, we highlight some of the key results from the submission. These are not new and have been presented thoroughly in the submitted paper.

| | Technique | Linear Classification (Tab. 1, 2) | Downstream *vs.*ImageNet supervised (Tab. 2) | Semi-supervised (Tab. 1) |
|---|---|---|---|---|
| Previous Best | Contrastive | MoCo-v2 or SimCLR | Better on detection. Worse on linear classification. | SimCLR |
| Ours | Clustering | $\Delta$ top-1 +**4.2**% ImageNet, +**4.6**% Places, +**9.7**% iNat18 | Better on both detection and linear classification | +**5.6**% top-1 ImageNet |

## High-level remarks

**R1Q1**: *"focuses too much on not using momentum and memory bank"* We thank **R1** for this feedback and will rewrite to de-emphasize the fact that we only use a single network. Our intention was not to challenge the momentum mechanism. Combining SwAV with a momentum encoder and/or a large memory bank are indeed interesting follow-ups.

**R2Q1 + R4**: *"fair comparison"* Comparisons to prior work are complicated as each work uses a different bag of tricks. In Tab.5, we make a best effort fair comparison (same data augmentation, num. epochs, batchsizes, etc). We observe in Tab.5 that clustering brings $+2$% compared to SimCLR and that multi-crop particularly improves clustering approaches.

**R1Q1**: *"with small batch, [MoCo is] go-to solution"* With batches of 256, SwAV reaches higher performance in half the time needed by MoCo: 72.0% after 102h (200ep) while MoCov2 reaches 71.1% after 212h (800ep). One epoch of MoCo is faster in wall clock time than one of SwAV, but MoCo needs more epochs for good downstream performance.

**R1Q7**: *"larger models"* Our paper shows large improvement over previous state of the art for all considered architectures, which suggests that SwAV does not overfit to R50 and can readily be applied to different models. The fact that BYOL (parallel work not available at submission time) has slightly better performance on large models does not imply that we suffer from poor scalability. Interestingly, our results follow a similar trend as supervised pretraining (Fig.2).

**R1Q7**: *"wall clock time for larger models"* We will add a "performance versus time" plot, similar to Fig.2 of the supplementary material, for large models. With R50w4, based on our implementation with 64 GPUs, SwAV gives 77.9% after 74.3h and 400ep while SimCLR reaches 76.8% after 130h and 1000ep (see Fig.7 of SimCLR paper).

**R1Q7**: *"lack of generality"* In Fig.4 we evaluate SwAV on random, uncurated images that have different properties from ImageNet and show that both our online clustering scheme and multi-crop augmentation work out of the box.

**R1Q1**: *"create artificial problems"* Although curation is a solution, training directly on uncurated data is an important research question that Fig.4 tries to address. Our intent was not to compare with MoCo so we will remove it from Fig.4.

## DeepCluster-v2 (DCv2)

**R2Q2**: *"how it is improved over initial version?"* We introduce explicit comparisons to k-means centroids, which increased stability, and leverage the training improvements from SimCLR. Full details are in supp. D. One goal of improving and using DCv2 as a baseline was to show the strong performance of clustering-based techniques. We train DCv2 in SwAV best setting (800 epochs - 8 crops) and obtain 75.2% top-1 accuracy on ImageNet.

**R1Q3 + R2Q2**: *"advantage of SwAV given DC-v2 is stronger?"* DCv2 performs comparably to SwAV. However, unlike SwAV, DCv2 is not online which makes it impractical for extremely large datasets. For billion scale trainings, as in MoCo, a single pass on the dataset is usually performed. DCv2 cannot be trained for only one epoch since it works by performing several passes on the dataset to regularly update centroids and cluster assignments for each image.

**R1Q3**: *"swapping mechanism does not seem important"* We respectfully disagree with this conclusion. DCv2 can be interpreted as a special case of our proposed swapping mechanism: swapping is done across epochs rather than within a batch. Given a crop of an image DCv2 predicts the assignment of another crop, which was obtained at the previous epoch. SwAV swaps assignments directly at the batch level and can thus work online.

**R1Q4**: *"efficiency comparison to DC-v2"* As discussed above, we will clarify the fact that k-means cost (12% of epoch time on ImageNet) is not the reason why DCv2 does not scale well.

## More evaluation experiments

**R3**: *"finetune results"* When finetuning R50 on Places and iNat18 we get 63.5% and 66.8% respectively, which is higher than training both from scratch and from ImageNet supervised model. We thank **R3** for the missing reference.

**R3**: *"semi-supervised learning with wider architectures"* Top-1 acc. on ImageNet – 1% labels: 56.5% (R50w2) / 58.7% (R50w4) - 10% labels: 72.6% (R50w2) / 74.5% (R50w4). We thank **R3** and will add the results in the paper.

**R1Q6 + R2Q3**: *"object detection is quite limited"* We agree that our gains on detection are limited and in the same ballpark as prior work. Yet, unlike prior work, our model outperforms supervised pre-training on both classification and detection tasks. As mentioned in supp. A.5, our VOC07 numbers are averaged over 5 runs.

## Implementation details and miscellaneous

**R1Q2**: *"area_range for the random_crop augmentation"* All details for reproducing SwAV trainings, including random_resized_crop parameters, are in supp. A.1 and A.2. Prior works have also tuned the random_crop: for example MoCo uses a scaling range of $(0.2, 1)$. For SwAV, using $(0.14, 1)$ gives $+0.2$% compared to $(0.08, 1)$ after 400 epochs.

**R1Q5**: *"data augmentation for linear probe"* SimCLR, MoCo and other prior works (all methods in Fig.2, Tab.2) use random crop augmentation when training linear probes on ImageNet, Places, iNat18. We follow their linear probe pipeline exactly to ensure our comparisons are fair. We do not use multi-crop for any of our evaluation.

**Misc.** We appreciate **R4**'s table re-organization suggestion. We agree with **R1** that mentioning RandAugment is not very informative and will remove it. We thank **R2** for their feedback and will clarify the use of the term "clustering": intuitively as the prototypes are used across different batches, SwAV "clusters" multiple instances to prototypes.

[Meta-Review · NeurIPS 2020]

The paper makes two incremental contributions in using online cluster assignments in self-supervised learning and using multiple crops in different resolutions for data augmentation. When these contributions are combined, decent gains in classification accuracy are obtained. The reviewers raise many issues with the current manuscript, including the discussion of momentum encoder, the discussion of existing clustering-based approaches, and the potential misuse of the term clustering. I ask the authors to incorporate all of these comments in the final version, but I believe the contributions even though incremental in nature, can benefit the fast growing field of self-supervised learning.